# Extending Stein's unbiased risk estimator to train deep denoisers with correlated pairs of noisy images

**Magauiya Zhussip**     **Shakarim Soltanayev**     **Se Young Chun**
Ulsan National Institute of Science and Technology (UNIST)
{mzhussip, shakarim, sychun}@unist.ac.kr

## Abstract

Recently, Stein's unbiased risk estimator (SURE) has been applied to unsupervised training of deep neural network Gaussian denoisers that outperformed classical non-deep learning based denoisers and yielded comparable performance to those trained with ground truth. While SURE requires only one noise realization per image for training, it does not take advantage of having multiple noise realizations per image when they are available (*e.g.*, two uncorrelated noise realizations per image for Noise2Noise). Here, we propose an extended SURE (eSURE) to train deep denoisers with correlated pairs of noise realizations per image and applied it to the case with two uncorrelated realizations per image to achieve better performance than SURE based method and comparable results to Noise2Noise. Then, we further investigated the case with imperfect ground truth (*i.e.*, mild noise in ground truth) that may be obtained considering painstaking, time-consuming, and even expensive processes of collecting ground truth images with multiple noisy images. For the case of generating noisy training data by adding synthetic noise to imperfect ground truth to yield correlated pairs of images, our proposed eSURE based training method outperformed conventional SURE based method as well as Noise2Noise. Code is available at `https://github.com/Magauiya/Extended_SURE`

## 1   Introduction

Powerful deep neural networks (DNNs) have been created and investigated for high-level computer vision tasks such as image classification [1, 2], object detection [3, 4], and semantic segmentation [5] as well as for low-level computer vision tasks such as image denoising [6, 7, 8, 9]. Initially, it was challenging for DNNs to outperform powerful classical denoisers such as BM3D [7]. However, recent works with DNNs proposed and demonstrated that it is possible for DNNs to outperform classical denoisers for synthetic Gaussian noise [8] as well as for real noise [10]. All aforementioned DNN based denoisers were trained in a supervised way with noiseless ground truth images.

Collecting high-quality noiseless images for training and evaluating DNN denoisers is challenging. Plotz and Roth collected high-quality benchmark data for denoising by averaging 19 independent noise realizations per one image [11]. It is a painstaking process to take tens of photos for one high-resolution image of static objects and to perform post-processing to compensate for lighting changes. It seems even more challenging, time-consuming, and even expensive to collect noiseless high-quality ground truth data for slowly moving objects (*e.g.*, animals, humans), for medical imaging, and for airborne hyper-spectral imaging. Even though it is possible to take 19 pictures, it may be inevitable that such a ground truth image contains mild noise if each picture is relatively noisy. For example, an average of 19 pictures contaminated with Gaussian noise of $\sigma = 25$ yields a picture with noise level of $\sigma = 5.74$ assuming temporal independence of noise. Thus, it seems desirable to have methods to deal with imperfect ground truth data (*i.e.*, mild noise in ground truth) and/or to train DNN denoisers without clean ground truth.

Recently, there have been several works on unsupervised training of DNN denoisers with noisy images only. Deep image prior (DIP) exploited the structure of a generator network and minimized the mean-squared error (MSE) between the output of a DNN and a given noisy image to denoise [12]. While DIP does not train the DNN, it requires to compute MSE minimization for each noisy image, which is slower than other DNN denoisers. Noise2Noise was proposed to train DNNs for image restoration with a set of two (independent) noise realizations per image in an unsupervised way for various noise models including Gaussian distribution and a wide range of applications including compressive sensing MR recovery [13]. There have been several self-supervised training works with only one noise realization per image. Stein's unbiased risk estimator (SURE) based training method for Gaussian denoisers was proposed to train DNNs with a set of a single noise realization per image [14]. SURE based training method has been extended to train DNNs with a set of undersampled compressive sensing measurements [15]. Noise2Void was proposed to train denoiser DNNs using a blind-spot network and yielded good performance, but it did not yield better performance than conventional methods such as BM3D especially for low noise levels [16]. Noise2Self was also proposed to train denoiser DNNs using J-invariant masks and yielded comparable performance to the network trained with ground truth for Hanzi dataset, but it yielded lower performance for CellNet dataset, possibly due to non-optimal selection of J-partition [17]. Laine *et al.* further improved Noise2Void by proposing blind-spot convolutional network architecture with restricted directional receptive field and Bayesian distribution prediction on output colors, yielding excellent performance comparable to the network trained with ground truth for Gaussian denoising [18]. Noise2Void has been extended to Noise2Boosting, an unbiased boosting estimator to train networks for more general range of applications such as super-resolution and accelerated MRI [19], and probabilistic Noise2Void using blind-spots to predict posterior probability of a pixel [20].

Both Noise2Noise and SURE based method outperformed classical denoising methods such as BM3D for synthetic Gaussian noise and they often yielded comparable performance to DNN denoisers that were trained with clean ground truth. Noise2Noise has demonstrated powerful performance in various image denoising and restoration tasks for zero-mean contaminations [13]. However, it required two independent noise realizations per image empirically, while there was no theoretical explanation on the relationship between two noise realizations. Thus, it is not clear if Noise2Noise can be used for the case with a single noise realization or for the case with imperfect ground truth data. Moreover, assuming slowly moving objects or slowly varying light conditions, there is a trade-off between low noise in ground truth and identical underlying true image over multiple realizations.

Even though SURE based training method is limited to Gaussian noise [14], it has a potential to be extended to more general noise models such as mixed Poisson-Gaussian model [21], exponential family [22] or non-parametric model [23]. It is also extended to unsupervised learning in inverse problems [24]. Moreover, several recent works on real noise denoising exploited a heteroscedastic Gaussian model $\mathbf{y} \sim \mathcal{N}(\mathbf{x}, \alpha + \beta \mathbf{x})$ with image generation procedures [10] or local AWGN with pixel-shuffle down-sampling [25]. Since SURE is a point-wise estimator and can deal with heteroscedastic Gaussian / local AWGN models, SURE based training scheme could be potentially useful for them. SURE could also be more robust to the noise-blur trade-off than Noise2Noise by using a single noise realization per image for slowly moving objects.

However, SURE based method is also limited since it does not take advantage of having multiple noise realizations per image when they are available (*e.g.*, two uncorrelated noise realizations per image for Noise2Noise, imperfect ground truth image with mild noise). In this paper, we address the following questions: 1) can Noise2Noise deal with correlated noise realizations and imperfect ground truth? 2) can SURE be extended to take advantage of having two uncorrelated noise realizations per image as in Noise2Noise? 3) can the extended SURE handle correlated noisy images and well utilize imperfect ground truth?

Here, we propose eSURE to training deep Gaussian denoisers with correlated pairs of noise realizations per image and applied it to the case with two uncorrelated realizations per image to achieve better performance than the original SURE based method and comparable results to Noise2Noise. Then, we further investigated the case of training Gaussian denoisers with imperfect ground truth (*i.e.*, mild noise in ground truth) by adding synthetic noise. For the case of adding noise to imperfect ground truth to yield correlated pairs of images, our proposed eSURE based training method outperformed conventional SURE based method as well as Noise2Noise.

Here is the summary on the contributions of this paper:

- Analyzing Noise2Noise theoretically and empirically for correlated pairs of noise realizations per image.
- Extending SURE to take advantage of having a pair of noise realizations per image for training unlike the conventional SURE that can take a single noise realization per image.
- Investigating eSURE that exploits two independent or correlated noise realizations per image as well as that can utilize imperfect ground truth with mild noise.

We will show that there is a clear theoretical link between Noise2Noise and eSURE in a limited case of Gaussian denoising. In fact, it turned out that Noise2Noise is a special case of eSURE for independent Gaussian noise in theory. However, while the performance of Noise2Noise was degraded with correlated pairs of noisy images, the performance of the proposed eSURE remains the same.

This paper is organized as follows: Section 2 briefly reviews SURE, Monte-Carlo SURE (MC-SURE) and SURE based denoiser training. Then, Section 3 revisits theoretical derivation of Noise2Noise training methods, proposes an extended version of MC-SURE (called eSURE) and shows a clear link between our eSURE and Noise2Noise. These are followed by experimental results to validate the effectiveness of our proposed unsupervised DNN training method in Section 4.1 for the case of having two independent realizations per image and for the case with imperfect ground truth. Finally, Section 5 draws a conclusion of this work.

## 2 Background

### 2.1 Stein's unbiased risk estimator (SURE)

Typically, Gaussian contaminated signal (or image) is modeled as a linear equation:

$$\mathbf{y} = \mathbf{x} + \mathbf{n} \tag{1}$$

where $\mathbf{x} \in \mathbb{R}^N$ is an unknown signal, $\mathbf{y} \in \mathbb{R}^N$ is a known measurement, $\mathbf{n} \in \mathbb{R}^N$ is an *i.i.d.* Gaussian noise such that $\mathbf{n} \sim \mathcal{N}(\mathbf{0}, \sigma^2 \mathbf{I})$, and $\mathbf{I}$ is an identity matrix. We denote $\mathbf{n} \sim \mathcal{N}(\mathbf{0}, \sigma^2 \mathbf{I})$ as $\mathbf{n} \sim \mathcal{N}_{0,\sigma^2}$.

In general, given an estmator $\mathbf{h}(\mathbf{y})$ of $\mathbf{x}$, the SURE has the following form:

$$\eta(\mathbf{h}(\mathbf{y})) = \frac{\|\mathbf{y} - \mathbf{h}(\mathbf{y})\|^2}{N} - \sigma^2 + \frac{2\sigma^2}{N} \sum_{i=1}^{N} \frac{\partial \mathbf{h}_i(\mathbf{y})}{\partial \mathbf{y}_i} \tag{2}$$

Assuming $\mathbf{x}$ to be deterministic signal (or image), the following theorem for (2) holds.

**Theorem 1.** *[26, 27] The random variable $\eta(\mathbf{h}(\mathbf{y}))$ is an unbiased estimator of*

$$\mathrm{MSE}(\mathbf{h}(\mathbf{y})) = \frac{1}{N}\|\mathbf{x} - \mathbf{h}(\mathbf{y})\|^2$$

*or*

$$\mathbb{E}_{\mathbf{n}\sim\mathcal{N}_{0,\sigma^2}} \left\{ \frac{\|\mathbf{x} - \mathbf{h}(\mathbf{y})\|^2}{N} \right\} = \mathbb{E}_{\mathbf{n}\sim\mathcal{N}_{0,\sigma^2}} \left\{ \eta(\mathbf{h}(\mathbf{y})) \right\} \tag{3}$$

*where $\mathbb{E}_{\mathbf{n}\sim\mathcal{N}_{0,\sigma^2}} \{\cdot\}$ is the expectation operator in terms of the random vector $\mathbf{n}$.*

Although (2) looks appealing in terms of optimizing parameters of an estimator $\mathbf{h}(\mathbf{y})$, the analytical solution for the last divergence term in (2) is limited only to some special cases such as the estimator $\mathbf{h}(\mathbf{y})$ to be non-local mean or linear filters [28, 29]. Thus, in order to utilize (2), one needs to find at least an approximate solution of the divergence term for more general cases.

### 2.2 Monte-Carlo SURE (MC-SURE)

A fast Monte-Carlo approximation of the divergence term has been developed by Ramani *et al.* in [30]. This method yielded accurate unbiased estimate of MSE for many denoising methods $\mathbf{h}(\mathbf{y})$.

**Theorem 2.** *[30] Let $\tilde{\mathbf{n}} \sim \mathcal{N}_{0,1} \in \mathbb{R}^N$ be independent of $\mathbf{n}$ or $\mathbf{y}$. Then,*

$$\sum_{i=1}^{K} \frac{\partial \mathbf{h}_i(\mathbf{y})}{\partial \mathbf{y}_i} = \lim_{\epsilon \to 0} \mathbb{E}_{\tilde{\mathbf{n}}} \left\{ \tilde{\mathbf{n}}^{\mathrm{T}} \left( \frac{\mathbf{h}(\mathbf{y} + \epsilon\tilde{\mathbf{n}}) - \mathbf{h}(\mathbf{y})}{\epsilon} \right) \right\} \tag{4}$$

*provided that $\mathbf{h}(\mathbf{y})$ admits a well-defined second-order Taylor expansion. If not, this is still valid in the weak sense provided that $\mathbf{h}(\mathbf{y})$ is tempered.*

Consequently, by applying Theorem 2 to the divergence term in (2), the divergence approximation of the denoiser $\mathbf{h}(\mathbf{y})$ will be:

$$\frac{1}{N}\sum_{i=1}^{N}\frac{\partial \mathbf{h}_i(\mathbf{y})}{\partial \mathbf{y}_i} \approx \frac{1}{\epsilon N}\tilde{\mathbf{n}}^{\mathrm{T}}\left(\mathbf{h}(\mathbf{y}+\epsilon\tilde{\mathbf{n}})-\mathbf{h}(\mathbf{y})\right), \tag{5}$$

where $\tilde{\mathbf{n}}^{\mathrm{T}}$ is a transposed $i.i.d$ Gaussian vector ($\tilde{\mathbf{n}} \sim \mathcal{N}_{0,1}$) and $\epsilon$ is a fixed small positive value.

### 2.3 SURE based deep denoiser training

Recently, SURE was used as a surrogate metric to minimize the MSE between the output of the DNNs and the ground truth for unsupervised training of DNN based Gaussian denoisers [14]. Specifically, MC-SURE allows DNN to learn large-scale weights by minimizing MC-SURE with no noiseless ground truth images for Gaussian denoising. The equation (2) with (4) was reformulated for the DNN $\mathbf{h}_{\boldsymbol{\theta}}(\cdot)$ as follows:

$$\eta(\boldsymbol{h}_{\boldsymbol{\theta}}(\mathbf{y})) = \frac{1}{M}\sum_{j=1}^{M}\left\{\|\mathbf{y}^{(j)}-\mathbf{h}_{\boldsymbol{\theta}}(\mathbf{y}^{(j)})\|^2 - N\sigma^2 \right. \tag{6}$$
$$\left. +\frac{2\sigma^2}{\epsilon}(\tilde{\mathbf{n}}^{(j)})^{\mathrm{t}}\left(\boldsymbol{h}_{\boldsymbol{\theta}}(\mathbf{y}^{(j)}+\epsilon\tilde{\mathbf{n}}^{(j)})-\boldsymbol{h}_{\boldsymbol{\theta}}(\mathbf{y}^{(j)})\right)\right\},$$

where $\boldsymbol{\theta}$ is the set of DNN denoiser parameters, $M$ is the size of mini-batch, $\epsilon$ is a small fixed positive constant, and $\tilde{\mathbf{n}}^{(j)}$ is a single realization from standard normal distribution for each training data $j$.

This approach has been demonstrated to yield state-of-the-art performance in denoising task with synthetic Gaussian noise, yielding comparable to or slightly worse qualitative and quantitative results than MSE-trained DNNs.

## 3 Methods

In this section, we first re-visit Noise2Noise method [13] and re-derive the method of Noise2Noise in a different approach from [13]. Then, we propose to extend the original SURE and MC-SURE to deal with a pair of correlated noisy images instead of a single noisy image and to use it for training deep learning based denoisers with pairs of correlated Gaussian noise realizations per image. We also show that Noise2Noise is a special case of our proposed extended SURE for Gaussian denoising. Lastly, we will demonstrate that our proposed method is more robust to correlated noise realization pairs in training data set compared to a Noise2Noise method. Our eSURE is especially useful for the case of using imperfect ground truth images with mild noise.

### 3.1 Revisiting Noise2Noise

The Noise2Noise method has been proposed to train DNNs for image processing only with noisy images where two noise realizations per image were required [13]. Its theoretical justification required zero-mean noise, but there was no clear assumption on independence or uncorrelated property of two realizations. However, two independent noise realizations per image were used empirically.

Assuming that the triplet $(\mathbf{x},\mathbf{y},\mathbf{z})$ follows a joint distribution and the expectation of two noise vectors $\mathbf{y}-\mathbf{x}, \mathbf{z}-\mathbf{x}$ are both zero vectors, the MSE for infinite data is as follows:

$$\begin{aligned}\mathbb{E}_{(\mathbf{x},\mathbf{y})}\left\{\|\mathbf{x}-\boldsymbol{h}_{\boldsymbol{\theta}}(\mathbf{y})\|^2\right\} &= \mathbb{E}_{\mathbf{x}}\left[\mathbb{E}_{(\mathbf{y},\mathbf{z})|\mathbf{x}}\left\{\|\mathbf{x}-\mathbf{z}+\mathbf{z}-\boldsymbol{h}_{\boldsymbol{\theta}}(\mathbf{y})\|^2|\mathbf{x}\right\}\right] \\ &= \mathbb{E}_{\mathbf{x}}\left[\mathbb{E}_{(\mathbf{y},\mathbf{z})|\mathbf{x}}\left\{\|\mathbf{z}-\boldsymbol{h}_{\boldsymbol{\theta}}(\mathbf{y})\|^2+2(\mathbf{z}-\mathbf{x})^T\boldsymbol{h}_{\boldsymbol{\theta}}(\mathbf{y})|\mathbf{x}\right\}\right]+const.\end{aligned} \tag{7}$$

Therefore, for a fixed $\mathbf{x}$, if $\mathbf{y}$ and $(\mathbf{z}-\mathbf{x})$ are uncorrelated or independent such that $(\mathbf{z}-\mathbf{x})$ has zero mean vector, then (7) is equivalent to the following Noise2Noise loss function in terms of $\boldsymbol{\theta}$:

$$\mathbb{E}_{(\mathbf{x},\mathbf{y},\mathbf{z})}\|\mathbf{z}-\boldsymbol{h}_{\boldsymbol{\theta}}(\mathbf{y})\|^2. \tag{8}$$

Consequently, the optimal network parameters $\boldsymbol{\theta}$ of a denoiser using (8) will yield the same solution as the MSE based training with clean ground truth. Noise2Noise achieved outstanding performance

in various image restoration tasks including Gaussian noise removal as far as there is the set of two noisy image pairs per one ground truth image [13].

Therefore, this analysis on Noise2Noise can now predict that if there are imperfect ground truth images $\tilde{\mathbf{x}}$ with a mild noise, then denoiser training with $\tilde{\mathbf{x}}$ plus additional synthetic noise may not be able to yield good performance comparable to the case using two independent noise realizations or using perfect ground truth data with additional synthetic noise possibly due to non-negligible non-zero term of $\mathbb{E}_{(\mathbf{x},\mathbf{y},\mathbf{z})}\left\{(\mathbf{z}-\mathbf{x})^T\boldsymbol{h_\theta}(\mathbf{y})\right\}$ in (7).

## 3.2 Extended SURE and MC-SURE

The original SURE in (2) works well with a single noise realization per image, but it can not take advantage of having multiple noise realizations per image just like Noise2Noise. Thus, we propose to extend the original SURE to be able to handle pairs of noisy images per ground truth image. The extended SURE (eSURE) can be formulated in the following way:

**Theorem 3.** *Let* $\mathbf{y_1} \sim \mathcal{N}(\mathbf{x}, \sigma_{\mathbf{y_1}}^2\mathbf{I})$ *be an imperfect ground-truth image,* $\mathbf{z} \sim \mathcal{N}(\mathbf{0}, \sigma_{\mathbf{z}}^2\mathbf{I})$ *is an AWGN, and* $\mathbf{y_2} \triangleq (\mathbf{y_1}+\mathbf{z}) \sim \mathcal{N}(\mathbf{x}, (\sigma_{\mathbf{y_1}}^2 + \sigma_{\mathbf{z}}^2)\mathbf{I})$ *is a noisy image. Then, the random variable* $\gamma(\boldsymbol{h_\theta}(\mathbf{y_2}), \mathbf{y_1})$ *is an unbiased estimator of MSE:*

$$\mathbb{E}_{\mathbf{y_2}}\left\{\frac{1}{N}\|\mathbf{x} - \boldsymbol{h_\theta}(\mathbf{y_2})\|^2\right\} = \mathbb{E}_{\mathbf{y_2}}\left\{\gamma(\boldsymbol{h_\theta}(\mathbf{y_2}), \mathbf{y_1})\right\}$$

*where* $\mathbf{y_1}$ *and* $\mathbf{z}$ *are independent (or uncorrelated) and*

$$\gamma(\boldsymbol{h_\theta}(\mathbf{y_2}), \mathbf{y_1}) = \frac{1}{N}\|\mathbf{y_1} - \boldsymbol{h_\theta}(\mathbf{y_2}))\|^2 - \sigma_{\mathbf{y_1}}^2 + \frac{2\sigma_{\mathbf{y_1}}^2}{N}\sum_{i=1}^{N}\frac{\partial \mathbf{h}_i(\mathbf{y_2})}{\partial(\mathbf{y_2})_i}. \tag{9}$$

Theorem 3 is developed for the general case where imperfect ground truth images with mild Gaussian noise are available and one needs to train the DNN for denoising images contaminated with larger noise level. Moreover, one can train DNN denoisers using the following Corollary of Theorem 3:

**Corollary.** *Given a noisy realization pairs of a clean image* $(\mathbf{y_3}, \mathbf{y_4})$ *from the same distribution* $\mathcal{N}(\mathbf{x}, \sigma_{\mathbf{y}}^2\mathbf{I})$*, we calculate less noisy image* $\mathbf{w} = \frac{1}{2}(\mathbf{y_3} + \mathbf{y_4}) \sim \mathcal{N}(\mathbf{x}, \frac{1}{2}\sigma_{\mathbf{y}}^2\mathbf{I})$*. Then, we add i.i.d. Gaussian noise* $\mathbf{z} \sim \mathcal{N}(\mathbf{0}, \frac{1}{2}\sigma_{\mathbf{y}}^2\mathbf{I})$ *to* $\mathbf{w}$*, so that* $\mathbf{v} = (\mathbf{w} + \mathbf{z}) \sim \mathcal{N}(\mathbf{x}, \sigma_{\mathbf{y}}^2\mathbf{I})$*. Finally, by applying Theorem 3 and replacing divergence term with its Monte-Carlo approximation (5), one can minimize extended MC-SURE with respect to* $\boldsymbol{\theta}$*:*

$$\gamma(\boldsymbol{h_\theta}(\mathbf{v}), \mathbf{w}) = \frac{1}{N}\|\mathbf{w} - \boldsymbol{h_\theta}(\mathbf{v})\|^2 - \frac{1}{2}\sigma_{\mathbf{y}}^2 + \frac{\sigma_{\mathbf{y}}^2}{\epsilon N}(\tilde{\mathbf{n}})^{\mathrm{T}}\left(\boldsymbol{h_\theta}(\mathbf{v} + \epsilon\tilde{\mathbf{n}}) - \boldsymbol{h_\theta}(\mathbf{v})\right). \tag{10}$$

For a training dataset of noisy $M$ pairs $\{(\boldsymbol{y_3}^{(1)}, \boldsymbol{y_4}^{(1)}), \cdots, (\boldsymbol{y_3}^{(M)}, \boldsymbol{y_4}^{(M)})\}$, we can generate $\{(\boldsymbol{w}^{(j)}, \boldsymbol{v}^{(j)})\}, j \in [1, M]$ to train deep learning based denoisers with proposed eSURE method. In simulations, our proposed method will show better performance compared to the original MC-SURE based training approach [14] for both grayscale and color image denoising. The proof of Theorem 3 and other details can be found in the supplementary material.

## 3.3 Link between eSURE and Noise2Noise

The eSURE framework that we proposed can be applied to a pair of uncorrelated Gaussian noisy images ($\mathbf{y} \sim \mathcal{N}(\mathbf{x}, \sigma_{\mathbf{y}}^2\mathbf{I})$ and $\mathbf{z} \sim \mathcal{N}(\mathbf{x}, \sigma_{\mathbf{z}}^2\mathbf{I})$). In that case, the divergence term vanishes leaving us the following expression:

$$\mathbb{E}_{\mathbf{z},\mathbf{y}}\left\{\gamma(\boldsymbol{h_\theta}(\mathbf{y}), \mathbf{z})\right\} = \mathbb{E}_{\mathbf{z},\mathbf{y}}\left\{\frac{\|\mathbf{z} - \boldsymbol{h_\theta}(\mathbf{y}))\|^2}{N}\right\} - \sigma_{\mathbf{z}}^2 \tag{11}$$

From the above expression, one clearly sees that the first term corresponds to the cost function of Noise2Noise for $i.i.d.$ Gaussian noise denoising case [13], while the second term $\sigma_{\mathbf{z}}^2$ is a constant.

Minimization of (11) with respect to a set of denoiser parameters $\boldsymbol{\theta}$ should give us the same solution for both Noise2Noise and our eSURE. Although it is not easy to notice the relationship between two different approaches from the first sight, it turns out that Noise2Noise is a special case of proposed extended MC-SURE based training method for $i.i.d.$ Gaussian denoising. A complete derivation can be found in the supplementary material.

# 4 Results

## 4.1 Experimental setup

We have conducted two experiments to evaluate our proposed methods. In the first experiment, we experimentally show that eSURE efficiently utilizes given two uncorrelated realizations per image to outperform SURE and is is a general case of Noise2Noise for $i.i.d.$ Gaussian noise. Second experiment aimed to investigate the effect of noise correlation for Noise2Noise and eSURE with imperfect ground truth. Our proposed method was compared with BM3D [31], DnCNN trained on MC-SURE [14], Noise2Noise [13] and DnCNN trained with MSE using noiseless ground truth data.

We used DnCNN [6, 8] as a deep denoising network for grayscale and RGB color images. DnCNN consists of 20 layers of CNN with batch normalization followed by ReLU as a non-linear function. For benchmark test images, we have chosen Berkeley's BSD-68[32] datasets, and widely used standard test images so called Set 12 [31]. All experiments were implemented on Tensorflow framework [33] and run on NVidia Titan X GPU.

It is worth to note that $\epsilon$ in (2) and (9) should be carefully chosen for stable training and high performance. As mentioned in [14] and [34], $\epsilon$ should be directly proportional to the noise standard deviation $\sigma$. Therefore, $\epsilon$ was fine tuned, so that for our proposed eSURE it was set to be $\epsilon = 1.6 \times 10^{-4} \times \sigma$. The results of ablation studies can be found in the supplements.

## 4.2 Case I: two uncorrelated noise realizations per one image

Given an uncorrelated two noisy realizations for each image in the training set, we trained DnCNNs with MC-SURE, Noise2Noise, and our proposed eSURE, respectively. More precisely, by following procedures described in DnCNN paper [8], we generated $128 \times 2,919$ patches with $50 \times 50$ size from BSD-400 [32] and produced two independent noisy patches (noise level range $\sigma \in [0 - 55]$) per clean patch. Using the aforementioned Corollary, we trained DnCNN with eSURE. It is worth to note that MC-SURE requires only a single noisy image per one ground truth in the training set compared to eSURE and Noise2Noise. Therefore, for a fair comparison, we concatenated both noisy datasets to train DnCNN-SURE with twice more data and denoted it as DnCNN-SURE*. More precisely, we treated two different realizations of the same image as different images. Once patches are extracted, we randomly permuted all patches for every epoch and optimized the network with them. DnCNN denoisers were trained for blind denoising with the noise level range of $\sigma \in [0 - 55]$ using Adam optimizer [35]. An initial learning rate was set to $10^{-3}$, which was dropped to $10^{-4}$ after 40 epochs and the network was further trained for 10 more epochs.

The performance of our approach along with the state-of- the-art methods was tabulated in Table 1. Our network training approach demonstrates almost identical quantitative results with Noise2Noise trained DnCNN (DnCNN-N2N) in both test sets. These results are consistent with our theoretical understandings such as 1) eSURE efficiently utilized two uncorrelated realizations compared to SURE*, 2) (11) holds for uncorrelated noisy training set and Noise2Noise is a special case of the extended MC-SURE. Moreover, quantitative analysis on BSD68 test set reveals that our eSURE is consistently better than conventional BM3D for about 0.5dB and outperforms DnCNN-SURE for about 0.15 dB in lower and higher noise cases. The performance gap between the proposed method, BM3D and DnCNN-SURE are still similar for Set12 test set. In addition, we can observe that minimizing MC-SURE with twice more dataset (DnCNN-SURE*) provides a little improvement, but not enough to reach DnCNN-eSURE and DnCNN-N2N.

In terms of visual comparison, our proposed eSURE method effectively removed noise from an image, while preserving texture and edges. In Figure 1, conventional BM3D yielded blurry results. A similar trend is observed for DnCNN-SURE where details of the denoised test image from BSD68 were not fully recovered (see Figure 1). Also one may be able to observe visually similar denoising performance for DnCNN-N2N and our method.

To sum up, it was experimentally shown that for two uncorrelated noise realizations of noisy training set, DnCNN-eSURE and DnCNN-N2N yield almost identical performance. Also, since eSURE uses less noisy data for training (see Corollary), it better approximates MSE and accordingly outperforms MC-SURE yielding results that are closer to MSE trained DnCNN.

Table 1: PSNR results of blind denoisers on BSD68 and Set12 datasets.

| Methods | BM3D | DnCNN-SURE | DnCNN-SURE* | DnCNN-N2N | DnCNN-eSURE | DnCNN-MSE |
|---|---|---|---|---|---|---|
| BSD-68 | | | | | | |
| $\sigma = 25$ | 28.56 | 28.92 | 29.00 | **29.08** | **29.08** | 29.20 |
| $\sigma = 50$ | 25.62 | 26.00 | 26.07 | 26.13 | **26.15** | 26.22 |
| Set 12 | | | | | | |
| $\sigma = 25$ | 29.97 | 30.04 | 30.13 | 30.30 | **30.31** | 30.42 |
| $\sigma = 50$ | 26.67 | 26.87 | 26.97 | **27.07** | **27.07** | 27.16 |
| Noisy dataset | - | 1 | 2 | 2 | 2 | $\infty$ |

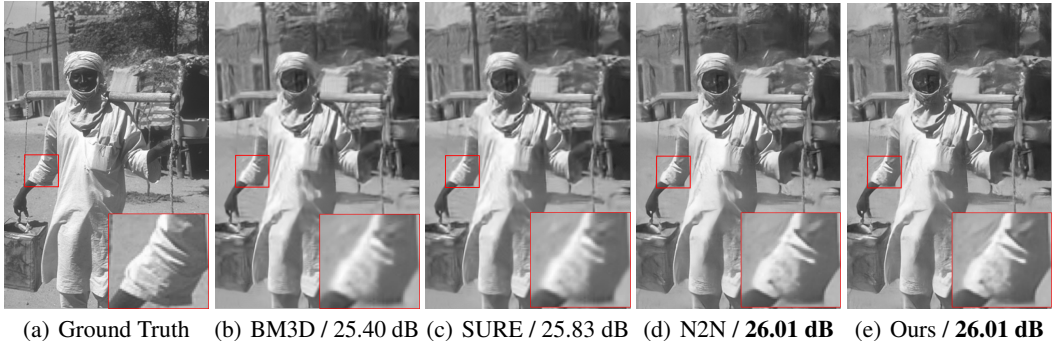

(a) Ground Truth  (b) BM3D / 25.40 dB  (c) SURE / 25.83 dB  (d) N2N / **26.01 dB**  (e) Ours / **26.01 dB**

Figure 1: Denoised test (BSD68) results of BM3D, DnCNN trained with various methods for $\sigma = 50$.

## 4.3   Case II: two correlated noise realizations per one image - imperfect ground truth

In many practical applications, collecting noiseless ground truth images is challenging, expensive, or even infeasible because of camera/object motion and long exposure time. Thus, we may have a limited number of photo shots of a particular scene and by averaging them, we have a ground truth where small amount of noise still remained. Adding synthetic noise to imperfect ground truth to produce noisy images for DNN training produces a dataset where the noise in noisy image may be correlated with the noise in the ground truth. In order to investigate how correlated noise affects our eSURE, Noise2Noise, and original SURE, we have conducted 2 experiments: DnCNN denoisers were trained for a fixed noise level for denoising (e.g. $\sigma_{noisy} = 25, 50$) on grayscale BSD-400 and for a blind noise ($\sigma_{noisy} \in [\sigma_{gt} - 55]$) on color BSD-432 [32].

We simulated the case with imperfect ground truth images (or slightly noisy ground truth), by adding synthetic Gaussian noise with $\sigma_{gt}$ to the noiseless clean oracle images. Consequently, noisy training images were generated by adding $i.i.d.$ Gaussian noise on the top of the imperfect ground truth dataset. Following the same procedures in Section 4.2, 128×2,919 patches with 50×50 size for grayscale and 128×2,019 patches for RGB case were generated. DnCNN denoisers were trained using Adam optimizer [35]. The initial learning rate was set to $10^{-3}$, which was dropped to $10^{-4}$ after 40 epochs and the network further trained for 10 more epochs.

Table 2 shows the performance of denoising methods trained for a fixed noise given a noisy ground-truth images with $\sigma_{gt} = \{1, 5, 10, 20\}$. The higher the $\sigma_{gt}$ is, the more correlated noise is in a training set. We notice that at low level of ground-truth noise ($\sigma_{gt} = 1$), both Noise2Noise (DnCNN-N2N) and eSURE (DnCNN-eSURE) yield the best PSNR results and even comparable to the DnCNN-MSE. However, as noise correlation gets severe ($\sigma_{gt} = 5, 10$), Noise2Noise fails to achieve high performance that is consistent with our theoretical derivation. In contrast, the proposed DnCNN eSURE produces the best quantitative results in a stable manner. Although, DnCNN-SURE is not susceptible to noise correlation, it still yielded worse performance than DnCNN-eSURE.

The experimental results for RGB color image denoising case are tabulated in Table 3. In this case, we observe the same performance degradation pattern of CDnCNN-N2N as the noise in ground truth image increases (more results in the supplementary material). The visual assessment of methods also demonstrates that eSURE trained CDnCNN was able to provide high-quality images with

Table 2: Results of denoising methods on BSD68 and Set 12 datasets (Performance in dB).

| | | BSD-68 | | | | | |
|---|---|---|---|---|---|---|---|
| $\sigma_{noisy}$ | | 25 | | | 50 | | |
| $\sigma_{gt}$ | 1 | 5 | 10 | 1 | 5 | 10 | 20 |
| BM3D | | 28.56 | | | 25.62 | | |
| DnCNN-SURE | 29.05 | 29.01 | 29.02 | 25.95 | 25.97 | 25.90 | 25.92 |
| DnCNN-N2N | 29.23 | 29.15 | 28.37 | **26.28** | 26.24 | 25.91 | 24.69 |
| DnCNN-eSURE | **29.23** | **29.23** | **29.21** | 26.27 | **26.24** | **26.27** | **26.25** |
| DnCNN-MSE | | 29.23 | | | 26.28 | | |
| | | Set 12 | | | | | |
| BM3D | | 29.97 | | | 26.67 | | |
| DnCNN-SURE | 30.23 | 30.19 | 30.19 | 26.77 | 26.85 | 26.73 | 26.74 |
| DnCNN-N2N | 30.41 | 30.39 | 29.46 | **27.28** | 27.20 | 27.08 | 25.39 |
| DnCNN-eSURE | **30.47** | **30.48** | **30.44** | 27.27 | **27.27** | **27.25** | **27.23** |
| DnCNN-MSE | | 30.47 | | | 27.28 | | |

Table 3: Results of denoising methods on RGB color BSD68 dataset(Performance in dB).

| | | RGB BSD-68 | | | | | |
|---|---|---|---|---|---|---|---|
| $\sigma_{noisy}$ | | 25 | | | 50 | | |
| $\sigma_{gt}$ | 1 | 5 | 10 | 1 | 5 | 10 | 20 |
| CBM3D | | 30.70 | | | 27.38 | | |
| CDnCNN-SURE | 30.97 | 30.98 | 30.99 | 27.63 | 27.68 | 27.64 | 27.63 |
| CDnCNN-N2N | 31.18 | 31.08 | 29.83 | 27.89 | 27.87 | 27.61 | 25.62 |
| CDnCNN-eSURE | **31.20** | **31.18** | **31.19** | **27.94** | **27.91** | **27.90** | **27.78** |
| CDnCNN-MSE | | 31.20 | | | 27.93 | | |

preserved texture and color. Moreover, we can see from Figure 2 that denoised output image of CBM3D is highly smoothed out and CDnCNN-N2N recovered image still have some noise, while our eSURE denoised image shows sharp edges with almost no noise. To imitate more practical case, we experimented with varied $\sigma_{gt} \in [1 - 10]$ to train denoisers for blind color image denoising ($\sigma_{noisy} \in [10.1 - 55]$) and tested on images with a fixed noise level (similar to Table 1) as shown in Table 4. This additional experiment yielded consistent results and our proposed eSURE method still outperforms other methods.

Table 4: Results of denoising methods on RGB color dataset with varied ground-truth noise (Performance in dB).

| Method | CBM3D | CDnCNN-SURE | CDnCNN-N2N | CDnCNN-eSURE | CDnCNN-MSE |
|---|---|---|---|---|---|
| $\sigma = 25$ | 30.70 | 30.92 | 30.73 | **31.15** | 31.20 |
| $\sigma = 50$ | 27.38 | 27.62 | 27.70 | **27.91** | 27.93 |

## 5   Conclusion

We have investigated properties of Noise2Noise and proposed eSURE that extended the original SURE to handle correlated pairs of noisy images efficiently for training DNN denoisers. For two uncorrelated noisy realizations per image, eSURE yielded better performance than SURE that implies efficient utilization of two uncorrelated noisy realizations as compared to SURE and SURE*. Our eSURE also yielded comparable performance to Noise2Noise that is consistent with our theoretical analysis. For two correlated noisy realizations per image or imperfect ground truth, eSURE still

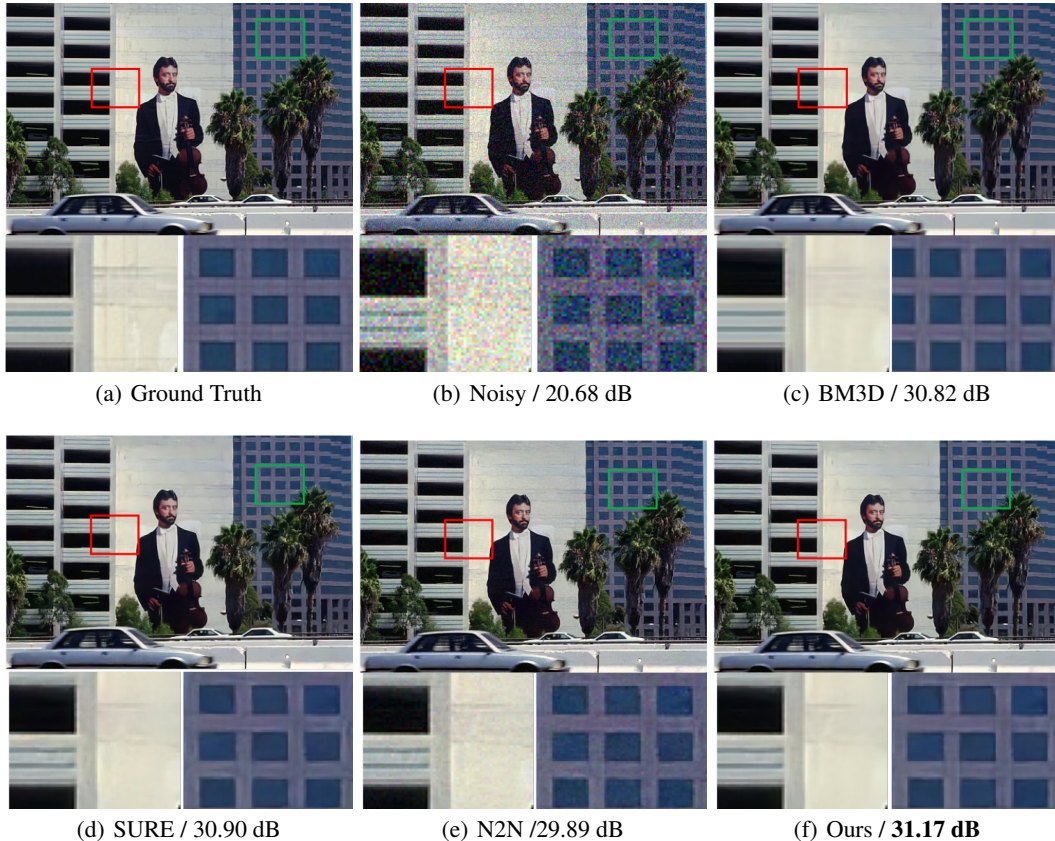

| (a) Ground Truth | (b) Noisy / 20.68 dB | (c) BM3D / 30.82 dB |
| (d) SURE / 30.90 dB | (e) N2N /29.89 dB | (f) Ours / **31.17 dB** |

Figure 2: CBM3D and CDnCNN results on test image from BSD-68 with noise $\sigma = 25$. CDnCNN trained with imperfect ground truth with $\sigma_{gt} = 10$ for blind noise denoising task using various approaches.

yielded the best performance among all compared methods such as BM3D, SURE, and Noise2Noise. However, Noise2Noise did not yield good performance with correlated noisy realizations as predicted based on our theoretical analysis.

# 6    Acknowledgments

This work was supported partly by Basic Science Research Program through the National Research Foundation of Korea(NRF) funded by the Ministry of Education(NRF-2017R1D1A1B05035810), the Technology Innovation Program or Industrial Strategic Technology Development Program (10077533, Development of robotic manipulation algorithm for grasping/assembling with the machine learning using visual and tactile sensing information) funded by the Ministry of Trade, Industry & Energy (MOTIE, Korea), and a grant of the Korea Health Technology R&D Project through the Korea Health Industry Development Institute (KHIDI), funded by the Ministry of Health & Welfare, Republic of Korea (grant number: HI18C0316).

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
