[Supplementary Material]

# Supplementary material: Extending Stein's unbiased risk estimator to train deep denoisers with correlated pairs of noisy images

**Magauiya Zhussip**      **Shakarim Soltanayev**      **Se Young Chun**
Ulsan National Institute of Science and Technology (UNIST)
{mzhussip, shakarim, sychun}@unist.ac.kr

## 1  Proof for Theorem 3

*Theorem 3:* Let $\mathbf{y_1} \sim \mathcal{N}(\mathbf{x}, \sigma_{\mathbf{y_1}}^2 \mathbf{I})$, $\mathbf{z} \sim \mathcal{N}(\mathbf{0}, \sigma_{\mathbf{z}}^2 \mathbf{I})$, and $\mathbf{y_2} \triangleq (\mathbf{y_1} + \mathbf{z}) \sim \mathcal{N}(\mathbf{x}, (\sigma_{\mathbf{y_2}}^2 + \sigma_{\mathbf{z}}^2)\mathbf{I})$. Then, the random variable $\gamma(\boldsymbol{h_\theta}(\mathbf{y_2}), \mathbf{y_1})$ is an unbiased estimator of MSE:

$$\mathbb{E}_{\mathbf{y_2}} \left\{ \frac{1}{N} \|\mathbf{x} - \boldsymbol{h_\theta}(\mathbf{y_2})\|^2 \right\} = \mathbb{E}_{\mathbf{y_2}} \left\{ \gamma(\boldsymbol{h_\theta}(\mathbf{y_2}), \mathbf{y_1}) \right\}$$

and equal to:

$$\gamma(\boldsymbol{h_\theta}(\mathbf{y_2}), \mathbf{y_1}) = \frac{1}{N} \|\mathbf{y_1} - \boldsymbol{h_\theta}(\mathbf{y_2}))\|^2 - \sigma_{\mathbf{y_1}}^2 + \frac{2\sigma_{\mathbf{y_1}}^2}{N} \sum_{i=1}^{N} \frac{\partial \mathbf{h}_i(\mathbf{y_2})}{\partial (\mathbf{y_2})_i} \tag{1}$$

*Proof:*

$$
\begin{aligned}
\mathbb{E}_{\mathbf{y_2}} \left\{ \frac{1}{N} \|\mathbf{x} - \boldsymbol{h_\theta}(\mathbf{y_2})\|^2 \right\} =& \mathbb{E}_{\mathbf{y_1},\mathbf{y_2}} \left\{ \frac{1}{N} \|\mathbf{x} - \mathbf{y_1} + \mathbf{y_1} - \boldsymbol{h_\theta}(\mathbf{y_2})\|^2 \right\} \\
=& \mathbb{E}_{\mathbf{y_1}} \left\{ \frac{1}{N} \|\mathbf{x} - \mathbf{y_1}\|^2 \right\} + \mathbb{E}_{\mathbf{y_1},\mathbf{y_2}} \left\{ \frac{1}{N} \|\mathbf{y_1} - \boldsymbol{h_\theta}(\mathbf{y_2})\|^2 \right\} \\
& + \frac{2}{N} \mathbb{E}_{\mathbf{y_1},\mathbf{y_2}} \left\{ (\mathbf{x} - \mathbf{y_1})^{\mathrm{T}}(\mathbf{y_1} - \boldsymbol{h_\theta}(\mathbf{y_2})) \right\} \\
=& \sigma_{\mathbf{y_1}}^2 + \mathbb{E}_{\mathbf{y_1},\mathbf{y_2}} \left\{ \frac{1}{N} \|\mathbf{y_1} - \boldsymbol{h_\theta}(\mathbf{y_2})\|^2 \right\} \\
& + \frac{2}{N} \mathbb{E}_{\mathbf{y_1},\mathbf{y_2}} \left\{ (\mathbf{x} - \mathbf{y_1})^{\mathrm{T}}(\mathbf{y_1} - \mathbf{x} + \mathbf{x} - \boldsymbol{h_\theta}(\mathbf{y_2})) \right\} \\
=& \mathbb{E}_{\mathbf{y_1},\mathbf{y_2}} \left\{ \frac{1}{N} \|\mathbf{y_1} - \boldsymbol{h_\theta}(\mathbf{y_2})\|^2 \right\} - \sigma_{\mathbf{y_1}}^2 \\
& + \frac{2}{N} \mathbb{E}_{\mathbf{y_1},\mathbf{y_2}} \left\{ (\mathbf{x} - \mathbf{y_1})^{\mathrm{T}}(\mathbf{x} - \boldsymbol{h_\theta}(\mathbf{y_2})) \right\} \\
=& \mathbb{E}_{\mathbf{y_1},\mathbf{y_2}} \left\{ \frac{1}{N} \|\mathbf{y_1} - \boldsymbol{h_\theta}(\mathbf{y_2})\|^2 \right\} - \sigma_{\mathbf{y_1}}^2 + \mathbf{0} \\
& + \frac{2}{N} \mathbb{E}_{\mathbf{y_1},\mathbf{y_2}} \left\{ (\mathbf{y_1} - \mathbf{x})^{\mathrm{T}} \boldsymbol{h_\theta}(\mathbf{y_2}) \right\}
\end{aligned}
\tag{2}
$$

In order to derive the last term in (2), let us first consider divergence for a pixel $i$.

$$
\begin{aligned}
\mathbb{E}_{(\mathbf{y_2})_i} \left\{ \frac{\partial \mathbf{h}_i(\mathbf{y_2})}{\partial (\mathbf{y_2})_i} \right\} &= \mathbb{E}_{(\mathbf{y_1},\mathbf{z})_i} \left\{ \frac{\partial \mathbf{h}_i(\mathbf{y_1}+\mathbf{z})}{\partial (\mathbf{y_1}+\mathbf{z})_i} \right\} \\
&= \iint_{-\infty}^{+\infty} \frac{\partial \mathbf{h}_i(\mathbf{y_1}+\mathbf{z})}{\partial (\mathbf{y_1}+\mathbf{z})_i} \phi_i(\mathbf{y_1})\phi_i(\mathbf{z}) d(\mathbf{y_1})_i d(\mathbf{z})_i \\
&= \int_{-\infty}^{+\infty} \left\{ \mathbf{h}_i(\mathbf{y_1}+\mathbf{z})\phi_i((\mathbf{y_1})) \Big|_{-\infty}^{+\infty} \right\} \phi_i(\mathbf{z}) d(\mathbf{z})_i \\
&\quad - \int_{-\infty}^{+\infty} \left\{ \mathbf{h}_i(\mathbf{y_1}+\mathbf{z})\frac{d\phi_i((\mathbf{y_1}))}{d(\mathbf{y_1})_i} \right\} \phi_i(\mathbf{z}) d(\mathbf{z})_i \\
&= \int_{-\infty}^{+\infty} \left\{ 0 + \int_{-\infty}^{+\infty} \frac{(\mathbf{y_1})_i - \mathbf{x}_i}{\sigma_{\mathbf{y_1}}^2} \mathbf{h}_i(\mathbf{y_1}+\mathbf{z})\phi_i(\mathbf{y_1}) d(\mathbf{y_1})_i \right\} \phi_i(\mathbf{z}) d(\mathbf{z})_i \\
&= \frac{1}{\sigma_{\mathbf{y_1}}^2} \iint_{-\infty}^{+\infty} (\mathbf{y_1})_i - \mathbf{x}_i) \mathbf{h}_i(\mathbf{y_1}+\mathbf{z})\phi_i(\mathbf{y_1})\phi_i(\mathbf{z}) d(\mathbf{y_1})_i d(\mathbf{z})_i \\
&= \frac{1}{\sigma_{\mathbf{y_1}}^2} \mathbb{E}_{(\mathbf{y_1},\mathbf{y_2})_i} \left\{ (\mathbf{y_1})_i - \mathbf{x}_i)\mathbf{h}_i(\mathbf{y_2}) \right\}
\end{aligned}
\tag{3}
$$

Here, $\phi_i(\mathbf{z})$ is a *pdf* of $\mathbf{z}_i$, while $(\mathbf{y_1})_i$ has a $\phi_i(\mathbf{y_1})$ pdf. Summation over each pixel gives us:

$$
\begin{aligned}
\sum_{i=1}^{N} \mathbb{E}_{(\mathbf{y_2})_i} \left\{ \frac{\partial \mathbf{h}_i(\mathbf{y_2})}{\partial (\mathbf{y_2})_i} \right\} &= \mathbb{E}_{\mathbf{y_2}} \left\{ \sum_{i=1}^{N} \frac{\partial \mathbf{h}_i(\mathbf{y_2})}{\partial (\mathbf{y_2})_i} \right\} \\
&= \frac{1}{\sigma_{\mathbf{y_1}}^2} \mathbb{E}_{\mathbf{y_1},\mathbf{y_2}} \left\{ (\mathbf{y_1} - \mathbf{x})^{\mathrm{T}} \boldsymbol{h_\theta}(\mathbf{y_2}) \right\}
\end{aligned}
\tag{4}
$$

Thus, by substituting the last correlation term in (2) with (4), we prove that:

$$
\mathbb{E}_{\mathbf{y_2}} \left\{ \frac{1}{N}\|\mathbf{x} - \boldsymbol{h_\theta}(\mathbf{y_2})\|^2 \right\} = \mathbb{E}_{\mathbf{y_1},\mathbf{y_2}} \left\{ \frac{1}{N}\|\mathbf{y_1} - \boldsymbol{h_\theta}(\mathbf{y_2}))\|^2 - \sigma_{\mathbf{y_1}}^2 + \frac{2\sigma_{\mathbf{y_1}}^2}{N} \sum_{i=1}^{N} \frac{\partial \mathbf{h}_i(\mathbf{y_2})}{\partial (\mathbf{y_2})_i} \right\}
\tag{5}
$$

## 2  Link between eSURE and Noise2Noise

Given a pair of uncorrelated noisy images ($\mathbf{y} \sim \mathcal{N}(\mathbf{x}, \sigma_{\mathbf{y}}^2)$ and $\mathbf{z} \sim \mathcal{N}(\mathbf{x}, \sigma_{\mathbf{z}}^2)$), the risk will be:

$$
\begin{aligned}
\mathbb{E}_{\mathbf{y}}\left\{\frac{1}{N}\|\mathbf{x} - \boldsymbol{h_\theta}(\mathbf{y})\|^2\right\} =&\, \mathbb{E}_{\mathbf{y},\mathbf{z}}\left\{\frac{1}{N}\|\mathbf{x} - \mathbf{z} + \mathbf{z} - \boldsymbol{h_\theta}(\mathbf{y})\|^2\right\} \\
=&\, \mathbb{E}_{\mathbf{y}}\left\{\frac{1}{N}\|\mathbf{x} - \mathbf{z}\|^2\right\} + \mathbb{E}_{\mathbf{y},\mathbf{z}}\left\{\frac{1}{N}\|\mathbf{z} - \boldsymbol{h_\theta}(\mathbf{y})\|^2\right\} \\
&+ \frac{2}{N}\mathbb{E}_{\mathbf{y},\mathbf{z}}\left\{(\mathbf{x} - \mathbf{z})^{\mathrm{T}}(\mathbf{z} - \boldsymbol{h_\theta}(\mathbf{y}))\right\} \\
=&\, \sigma_{\mathbf{z}}^2 + \mathbb{E}_{\mathbf{y},\mathbf{z}}\left\{\frac{1}{N}\|\mathbf{z} - \boldsymbol{h_\theta}(\mathbf{y})\|^2\right\} \\
&+ \frac{2}{N}\mathbb{E}_{\mathbf{y},\mathbf{z}}\left\{(\mathbf{x} - \mathbf{z})^{\mathrm{T}}(\mathbf{z} - \mathbf{x} + \mathbf{x} - \boldsymbol{h_\theta}(\mathbf{y}))\right\} \\
=&\, \mathbb{E}_{\mathbf{y},\mathbf{z}}\left\{\frac{1}{N}\|\mathbf{z} - \boldsymbol{h_\theta}(\mathbf{y})\|^2\right\} - \sigma_{\mathbf{z}}^2 \\
&+ \frac{2}{N}\mathbb{E}_{\mathbf{y},\mathbf{z}}\left\{(\mathbf{x} - \mathbf{z})^{\mathrm{T}}(\mathbf{x} - \boldsymbol{h_\theta}(\mathbf{y}))\right\} \\
=&\, \mathbb{E}_{\mathbf{y},\mathbf{z}}\left\{\frac{1}{N}\|\mathbf{z} - \boldsymbol{h_\theta}(\mathbf{y})\|^2\right\} - \sigma_{\mathbf{z}}^2 + \mathbf{0} + \frac{2}{N}\mathbb{E}_{\mathbf{y},\mathbf{z}}\left\{(\mathbf{z} - \mathbf{x})^{\mathrm{T}}\boldsymbol{h_\theta}(\mathbf{y})\right\} \\
=&\, \mathbb{E}_{\mathbf{y},\mathbf{z}}\left\{\frac{1}{N}\|\mathbf{z} - \boldsymbol{h_\theta}(\mathbf{y})\|^2\right\} - \sigma_{\mathbf{z}}^2
\end{aligned}
\tag{6}
$$

Since, both noisy images are uncorrelated, the term $\frac{2}{N}\mathbb{E}_{\mathbf{y},\mathbf{z}}\left\{(\mathbf{z} - \mathbf{x})^{\mathrm{T}}\boldsymbol{h_\theta}(\mathbf{y})\right\}$ in (6) vanishes.

## 3  Tuning $\epsilon$ for accurate eSURE estimation

Proposed method as well as MC-SURE depend on $\epsilon$ value and Soltanayev *et al.* found that $\epsilon$ is proportional to $\sigma$. Thus, we trained DnCNN-eSURE for a fixed noise ($\sigma_{gt} = 0$, $\sigma_{noisy} = 25$ and 50) with different $\epsilon$ values ranging from $10^{-3}$ to $10^{-2}$ and discovered that the ratio $\frac{\epsilon}{\sigma}$ is constant ($1.6 \times 10^{-3}$) for the best performances of networks. Consequently, for all experiments in our work, we set $\epsilon = 1.6 \times 10^{-4} \times \sigma$.

Table 1: Results of DnCNN-eSURE on BSD68 dataset for different $\epsilon$ values (Performance in dB).

| $\epsilon \times 10^{-3}$ | 1.0 | 2.5 | 4.0 | 5.0 | 6.0 |
|---|---|---|---|---|---|
| $\sigma = 25$ | 28.88 | 29.00 | **29.11** | 29.07 | 29.04 |
| $\epsilon \times 10^{-3}$ | 1.0 | 5.0 | 7.5 | 9.0 | 13.0 |
| $\sigma = 50$ | 25.87 | 26.02 | **26.07** | 26.03 | 25.72 |

## 4  Additional Simulation results

We also experimented with blind denoisers trained on grayscale BSD-400 and found that Noise2Noise is more susceptible to correlated noise in case if imperfect ground truth images were given (see Table 2). In order to see the performance degradation of Noise2Noise, we took DnCNN-N2N trained using different noise level of the imperfect ground-truth dataset and tested them on "Barbara" test image with noise $\sigma = 25$ (see Figure 2). In addition, we provided results for different cases for visual assessment.

Table 2: Results of denoising methods on BSD68 and Set 12 datasets (Performance in dB).

| | \multicolumn BSD-68 | | | | | |
|---|---|---|---|---|---|---|
| $\sigma_{noisy}$ | 25 | | | 50 | | |
| $\sigma_{gt}$ | 1 | 5 | 10 | 1 | 5 | 10 |
| BM3D | | 28.56 | | | 25.62 | |
| DnCNN-SURE | 28.92 | 28.92 | 28.88 | 25.95 | 25.97 | 25.95 |
| DnCNN-N2N | 29.10 | 29.02 | 28.22 | 26.16 | 26.14 | 25.99 |
| DnCNN-eSURE | **29.11** | **29.11** | **29.12** | **26.17** | **26.16** | **26.18** |
| DnCNN-MSE | | 29.23 | | | 26.28 | |
| | Set 12 | | | | | |
| BM3D | | 29.97 | | | 26.67 | |
| DnCNN-SURE | 30.03 | 30.02 | 30.00 | 26.75 | 26.79 | 26.78 |
| DnCNN-N2N | 30.33 | 30.21 | 29.24 | 27.14 | 27.11 | 26.91 |
| DnCNN-eSURE | **30.34** | **30.33** | **30.33** | **27.15** | **27.13** | **27.15** |
| DnCNN-MSE | | 30.42 | | | 27.16 | |

(a) Ground Truth     (b) Noisy / 20.28 dB     (c) BM3D / 29.86 dB

(d) SURE / 29.87 dB     (e) Noise2Noise / 29.12 dB     (f) Ours / **30.13 dB**

Figure 1: BM3D and DnCNN results on "Boat" test image from BSD-68 with noise $\sigma = 25$. DnCNN trained with imperfect ground truth with $\sigma_{gt} = 10$ for blind noise denoising task using various approaches..

(a) N2N {$\sigma_{gt} = 1$} / 32.35 dB   (b) N2N {$\sigma_{gt} = 5$} / 32.14 dB   (c) N2N {$\sigma_{gt} = 10$} / 30.73 dB   (d) N2N {$\sigma_{gt} = 20$} / 23.64 dB

Figure 2: Degradation of Noise2Noise performance as the noise level of an imperfect ground-truth in training set increases and hence noise correlation increases. DnCNN-N2N was tested on 'Barbara' test image with $\sigma = 25$ noise standard deviation.

(a) Ground Truth          (b) Noisy / 14.78 dB          (c) BM3D / 29.77 dB

(d) SURE / 29.73 dB       (e) Noise2Noise /29.97 dB     (f) Ours / **30.35 dB**

Figure 3: BM3D and DnCNN results on test image from BSD-68 with noise $\sigma = 50$. DnCNN trained with imperfect ground truth with $\sigma_{gt} = 10$ for specific noise denoising task using various approaches.

(a) Ground Truth

(b) Noisy / 14.69 dB

(c) BM3D / 27.66 dB

(d) SURE / 28.02 dB

(e) Noise2Noise /26.07 dB

(f) Ours / **28.22 dB**

Figure 4: CBM3D and CDnCNN results on test image from BSD-68 with noise $\sigma = 50$. CDnCNN trained with imperfect ground truth with $\sigma_{gt} = 10$ for blind noise denoising task using various approaches.