[Reviews · NeurIPS 2019]

Reviewer 1



Overall, the proposed work is interesting but the work is not clearly presented. The questions are: 1. The first experiments showed better results by eSURE than those of SURE. How is SURE trained using "twice more data"? Is it by increasing the batch size by 2? It would be good to see the results by twice batch size: 1) with two realizations of the same image; 2) different images. 2. For the second experiments with imperfect ground truth, what is the sigma in the training? The eSURE requires sigma in Eq. (10). Is that sigma_noisy in Tables 2 and 3? Does eSURE need to know sigma_gt? 3. Line 244, it seems that each network is trained for each noise level. So 7 DnCNN-eSURE networks were trained for Table 2 or 3? This is not practical and does not make sense. Normally, sigma_gt is unknown and may vary in the same dataset. It is reasonable to see one network for all different noise levels, like that in Table 1 for blind denoising.

Reviewer 2



Summary: The authors demonstrate that SURE and Noise2Noise, two methods that allow training a denoiser without clean ground truth can be viewed in the same theoretical framework. They show theoretically that Noise2Noise is not generally applicable in cases were the noise in the two images correlated, as it is e.g. the case when training is done with ground truth created by averaging noisy images. They present a novel method (eSURE) as a solution and show that Noise2Noise is a special case of eSURE. While eSURE in general is superior to Noise2Noise in the sense that it can be applied with correlated noise, it also yields slightly better results than vanilla SURE. The method is evaluated on the standard denoising datasets and is compared to sensible baselines (Noise2Noise, vanilla SURE, and standard supervised training). Its performance is in agreement with the theory. Originality: + The paper presents a highly original theory and derives a novel training method from it. Quality: + The paper seems theoretically and technically sound. Clarity: + The paper is well structured. - The clarity of the theory might be improved by being more explicit in some explanations. e.g.: Theorem 3 would be easier to understand if it would be explicitly stated (at least this is how I read it) that y_1 could e.g. correspond to a ground truth with remaining noise and that y_2 could correspond to an input image with correlated noise. Similarly for the Eq. 7, if the vectors (y-x) and (z-x) were to be introduced mentioning explicitly what they correspond to, the theory might be easier to grasp. Significance: I believe the paper is highly significant and can further our understanding of machine learning from noisy data. The findings are fundamental and important. Final Recommendation: Considering the principled original approach and the high significance, I firmly recommend to accept the paper. ------------------------------------------ Post Rebuttal: I feel that my concerns and suggestions have been adequately addressed and I will stick with my initial rating.

Reviewer 3



This paper is well written and the contribution is clearly descibed. My main concern is whether the assumption of correlated pairs of noisy images is helpful for real-world image denoising. I expect that the authors can give us some examples and initial results to illustrate this point for real application tasks. My second concern is that the proposed method is derived based on AWGN, while the real noise may be far different from such assumption and can be both signal-dependent and spatially variant. [1] Toward convolutional blind denoising of real photographs, CVPR 2019. [2] When AWGN-based Denoiser Meets Real Noises, Arxiv 2019. It seems that the proposed method is sensitive to the paramter \epison (see Lines 202~203). Thus, it is suggested to conduct an ablation study to see the effect of \epison.

[Author Response · NeurIPS 2019]

**To Reviewer 1** For "twice more data" case (called DnCNN-SURE*), we treated two different realizations of the same
image as different images. Once patches are extracted, we randomly permuted all patches for every epoch and optimized
the network with them. Your requests are interesting and we wish to show you the results for your requests, but due
to our hardware limitation, training was not working with two times of our current batch size. However, in light of
stochastic optimization, we argue that our random permutation does not seem problematic. We wanted to keep our
current settings since we would like to use the same optimization setting for all methods (so, needed fixed batch size).

For the imperfect ground truth experiments, SURE requires known $\sigma_{noisy}$, but our eSURE requires known $\sigma_{gt}$
(otherwise, they are not working). In Tables 2, 3, both $\sigma_{noisy}$ and $\sigma_{gt}$ are described in the second and third rows,
respectively. In practice, there are several methods to accurately estimate noise levels even for real noise [11], [A, B].

[A] S Pyatykh et al. IEEE Trans Im Proc 22(2):687-99 2012 [B] A Abdelhamed et al. CVPR 1692-1700 2018.

Even though Tables 2, 3 may not be practical, they did show the impact of our proposed method for different levels of
$\sigma_{gt}$ explicitly. However, your comment is correct for practical sense, so we did train one deep neural network with
varying $\sigma_{gt} \in [1 - 10]$ and $\sigma_{noisy} \in [10.1 - 55]$ for blind color image denoising and tested on images with a fixed
noise level (just like Table 1) as shown in the below table. This new experiment yielded consistent results and our
proposed eSURE method still outperforms other methods. Hope that the results of this table are convincing to you.

| Methods | CBM3D | DnCNN-SURE | DnCNN-N2N | DnCNN-eSURE | DnCNN-MSE |
|---|---|---|---|---|---|
| $\sigma = 25$ | 30.70 | 30.92 | 30.73 | **31.15** | 31.20 |
| $\sigma = 50$ | 27.38 | 27.62 | 27.70 | **27.91** | 27.93 |

**To Reviewer 2** It is indeed a good idea to be more explicit in some explanations for easier understanding. We will
modify some descriptions to reflect your comments such as $\mathbf{y_1}, \mathbf{y_2}$ in Theorem 3 and $\mathbf{y} - \mathbf{x}, \mathbf{z} - \mathbf{x}$ in Eq. 7 that are two
noise vectors with zero mean. We thank you for all suggested literature and we will cite them with proper discussions.

**To Reviewer 3** We briefly mentioned some cases with correlated pairs of noisy images in the introduction, but here are
more explanations on possible cases for correlated pairs of noisy images. We must say that our work may NOT be a
direct solution to the following cases, but we do believe that our work should have an important impact on them.

1) **Imperfect noisy ground truth images.** There exist many ground truth datasets available for denoiser training, but
many of them are not completely noise-free [H]. There are also cases where obtaining noise-free ground truth images is
challenging such as satellite imaging and/or medical imaging (CT, MRI). For example, high radiation dose is required
for clean ground truth of X-ray CT and it often takes tens of hours to obtain one volume of high resolution MR. Thus,
dealing with imperfect noisy ground truth seems potentially practical for these applications. One may argue that two
noise realizations per image for Noise2Noise could be obtained for the above cases, but note that it requires two times
larger storage which may not be favorable for the cases such as satellites with limited resources or medical devices
collecting large image volumes. Our work showed example experiments for them.

2) **Applications with spatio-temporal resolution trade-off.** As shown in our work, it is always beneficial to reduce
the noise level of input images for denoisers for improved performance. Now let us look at one example. Assume that
we have a sequence of four images $\mathbf{y_1} \sim \mathbf{y_4}$ per 100 ms with i.i.d Gaussian noise. If one decided to sacrifice temporal
resolution by temporal averaging, then there are several ways to do so. For Noise2Noise, to maintain the independence,
one may perform moving average like $(\mathbf{y_1} + \mathbf{y_2})/2$ and $(\mathbf{y_3} + \mathbf{y_4})/2$. However, for our eSURE based method, one
may have another option for moving average like $(\mathbf{y_1} + \mathbf{y_2})/2$ and $(\mathbf{y_2} + \mathbf{y_3})/2$. The first averaged images are over
100 ms, but the second averages images are over 75 ms. Thus, we argue that our proposed method provides more
flexibility than Noise2Noise for the cases with spatio-temporal resolution trade-off. The concept of temporal averaging
for better image quality is actually not just our example. Temporal averaging has been often used for some applications
in dynamic imaging [C, D]. Recently, there was a work on high speed camera to violate temporal independence [E].

3) Lastly, we hope that our work with more flexibility will motivate more applications with correlated pairs.

[C] KA Mohan et al. IEEE Trans Comp Imag 1(2):96-111 2015 [D] E Gravier et al. IEEE Trans Im Proc 16(4):932-42
2007 [E] Y Lu et al. Phys Rev Lett 122(19):193904 2019

For AWGN concern, first of all, SURE with AWGN has been extended to other noise models such as exponential
family [16] and nonparametric models [17], so it is potentially possible that SURE based unsupervised learning could
be extended for other noise models. Secondly, several recent works on real noise denoising exploited a heteroscedastic
Gaussian model $\mathbf{y} \sim \mathcal{N}(\mathbf{x}, \alpha + \beta\mathbf{x})$ with image generation procedures [F, G] or local AWGN with pixel-shuffle
down-sampling [H]. Since our proposed method can deal with heteroscedastic Gaussian / local AWGN models (SURE
is a point-wise estimator), our method could be potentially useful for them. This discussion with [F-H] will be added.

[F] S Guo et al. CVPR 1712-22 2019 [G] T Brooks et al. CVPR 11036-44 2019 [H] Y Zhou et al. ArXiv 2019

Lastly, we will report our ablation study to find the relationship between $\epsilon$ and $\sigma$ that was shown in Line 203.

[Meta-Review · NeurIPS 2019]

The reviewers agree that this submission represents an important contribution to the field. Please be sure to carefully review and address the concerns of all reviewers in the revision.